# ACSL3 and ACSL4, Distinct Roles in Ferroptosis and Cancers

**DOI:** 10.3390/cancers14235896

**Published:** 2022-11-29

**Authors:** Yufei Yang, Ting Zhu, Xu Wang, Fen Xiong, Zhangmin Hu, Xuehan Qiao, Xiao Yuan, Deqiang Wang

**Affiliations:** 1Department of Medical Oncology, Affiliated Hospital of Jiangsu University, Zhenjiang 212001, China; 2Institute of Oncology, Affiliated Hospital of Jiangsu University, Zhenjiang 212001, China; 3Institute of Digestive Diseases, Jiangsu University, Zhenjiang 212001, China

**Keywords:** ACSL3, ACSL4, ferroptosis, cancer

## Abstract

**Simple Summary:**

The dysregulation of ACSL3 and ACSL4, which belong to the long-chain fatty acyl CoA synthetase family (ACSLs), affects the behavior of various cancer cells. This review presents their distinct roles in ferroptosis and a summary of the double-edged effects of different cancers. Ferroptosis is a unique type of regulated cell death process which is caused by lipid peroxidation. Targeting the molecular mechanisms of ACSL3 and ACSL4 may provide more therapies for cancer treatments.

**Abstract:**

The long-chain fatty acyl CoA synthetase (ACSLs) family of enzymes contributes significantly to lipid metabolism and produces acyl-coenzyme A by catalyzing fatty acid oxidation. The dysregulation of ACSL3 and ACSL4, which belong to the five isoforms of ACSLs, plays a key role in cancer initiation, development, metastasis, and tumor immunity and may provide several possible therapeutic strategies. Moreover, ACSL3 and ACSL4 are crucial for ferroptosis, a non-apoptotic cell death triggered by the accumulation of membrane lipid peroxides due to iron overload. Here, we present a summary of the current knowledge on ACSL3 and ACSL4 and their functions in various cancers. Research on the molecular mechanisms involved in the regulation of ferroptosis is critical to developing targeted therapies for cancer.

## 1. Introduction

Fatty acid metabolism provides energy for cell metabolism and growth by degrading fatty acids. Dysregulated fatty acid metabolism drives lipid synthesis and deposition and is associated with several cancers [1]. Fatty acid metabolism is regulated by various enzymes [2]. The long-chain fatty acyl CoA synthetase family (ACSLs) contains five isoforms identified as ACSL1, 3, 4, 5, and 6. They convert free long-chain fatty acids into fatty acyl-CoA esters and play a dominant role in both anabolic (fatty acid synthesis and lipogenesis) and catabolic pathways (lipolysis and fatty acid β-oxidation), although they have distinct substrate specificity, subcellular localization, and tissue distribution [3]. ACSLs affect the behavior of malignant cells, including proliferation, migration, invasion, apoptosis, and drug resistance.

Ferroptosis was first coined by Dr. Brent R. Stockwell in 2012 [4]. Ferroptosis is a unique type of regulated cell death (RCD) process driven by the accumulation of membrane lipid peroxides (LPO) depending on iron overload, which can be regulated by specific pathways or cell metabolism. 

Ferroptosis involves cell metabolism, regulations of reactive oxygen species (ROS), and iron metabolism. To trigger ferroptosis, specific lipids must undergo peroxidation, and the natural defense mechanisms that prevent the accumulation of peroxidized lipids must be compromised. ACSLs play an important role in tissue cell metabolism, and different isoforms have different tissue distributions and substrate preferences, which regulate different intracellular lipid compositions. Among these five isoforms, ACSL3 and ACSL4 have been shown to participate in ferroptosis. In addition, ACSL4 is a positive regulator in ferroptosis, whereas ACSL3 contributes to cancer cells acquiring ferroptosis resistance [5].

Resisting cell death is one of the hallmarks of cancers [6]. Targeting ferroptosis promoters or inhibitors can be a potential and effective therapy for cancers, especially tumors that are susceptible to chemotherapy resistance. Moreover, immunotherapies have become a trend in cancer therapies. Since ferroptosis is tightly linked to lipid peroxidation, it is uncertain how exogenous lipid levels affect cell death sensitivity. In this review, we clarify the ACSLs family and the heterogeneity and functions of ACSL3 and ACSL4 in ferroptosis. Next, we discuss the mechanisms of the different regulation outcomes. We also provide the current understanding of and research on ACSL3 and ACSL4 in multiple cancers. Furthermore, we present a perspective on cancer therapies targeting or combining ACSL3 and ACSL4 in ferroptosis.

## 2. ACSL3 and ACSL4

During lipid metabolism, the activation of fatty acids by esterification with coenzyme A is indispensable. ACSLs catalyze this reaction which is the first step in the utilization of fatty acids in mammals [7] (Figure 1). The main process of catalysis and the metabolic pathways of long-chain fatty acids are described in Figure 2 [8,9]. Researchers have already found that ACSL3 and ACSL4 are involved in ferroptosis [10]. However, information is limited regarding the specific roles of ACSL3 and 4 in ferroptosis.

ACSL3 is highly expressed in the brain, testes, and skeletal muscle, and preferentially utilizes myristate, arachidonate, and eicosapentaenoate as substrates. ACSL3 mainly resides in the Golgi apparatus, endoplasmic reticulum (ER), peroxisomes, and mitochondria [11]. ACSL3 is a lipid droplet (LD)-associated protein. Most cells have a large amount of ACSL3 on the surface of LDs, and this protein has a number of pathophysiological functions in LD formation, autophagy control, and cellular ferroptosis [12]. It participates in several pathological processes, including LD production, autophagy control, and cellular ferroptosis.

ACSL4 is mainly located in the ER, mitochondria, plasma membrane, and peroxisome, and is enriched in the adrenal gland, ovary, testis, and brain tissues [3]. Microarray analysis using a genome-wide CRISPR-based genetic screening system and iron death-resistant cell lines revealed that ACSL4 is required for the induction of ferroptosis through the accumulation of oxidized cell membrane phospholipids (PLs) [5,13].

## 3. ACSL3 and ACSL4 in Ferroptosis

Ferroptosis is characterized by excessive accumulation and failure to eliminate iron-dependent lethal toxic lipid ROS, thereby initiating cell death. As a type of RCD, ferroptosis is either directed by intracellular signals or induced by external factors, leading to the initiation of suicidal protective strategies. It is considered a highly complicated and strictly orchestrated process that is regulated by a series of signals from distinct cell organelles, such as mitochondria, ER, and lysosomes [14].

Morphologically, distinct from other types of RCD such as apoptosis, necroptosis, and pyroptosis, ferroptosis shows swelling of cells, formation of pores in the cell membrane, smaller mitochondria, reduced mitochondrial cristae, increased mitochondrial membrane density, normal-sized nuclei and non-cohesive chromatin [15].

Ferroptosis is initiated by the accumulation of iron, lipid peroxidation, and subsequent plasma membrane rupture. Lipid peroxidation is a free radical-driven reaction that primarily affects polyunsaturated fatty acids (PUFA) in the cell membrane and undergoes three stages including initiation, propagation, and termination. Finally, lipid peroxidation produces lipid hydroperoxides [14]. Iron can favor tumor incidence and growth, as it is an important nutrient for cell proliferation and a key cofactor of metabolic enzymes [16]. Cellular or organelle membranes are particularly susceptible to lipid peroxidation because of their high PUFA content [17]. Cystatin-GSH-GPX4, CoQ10-FSP1, and GCH1-BH4-DHFR can effectively resist ferroptosis by inhibiting lipid peroxidation [18] (Figure 3).

Fatty acids belong to three categories: saturated fatty acids (no double bond), monounsaturated fatty acids (MUFA), and PUFA. Esterified PUFA is mainly affected by lipid peroxidation during ferroptosis. It is essential to consume sufficient amounts of unsaturated fatty acids to facilitate ferroptosis [19]. How PUFAs are oxidized to produce peroxynitrite and trigger iron death should be further investigated. PUFA and MUFA have opposite roles in ferroptosis because of their differential susceptibility to oxidation [18]. Ferroptosis is more susceptible to ferroptosis in the presence of increased PUFA synthesis, which is mainly controlled by ACSL4 and recombinant lysophosphatidylcholine acyltransferase 3. ACSL3 and stearyl coenzyme A dehydrogenase-1 (SCD-1) inhibit PUFA-dependent ferroptosis, respectively [10,20].

### 3.1. ACSL3 in Ferroptosis

ACSL3 may be involved in intracellular lipid metabolism by promoting LD formation and maturation. LDs are commonly found in the organelles of various cells, and they store neutral lipids, such as TAG and cholesteryl ester. ACSL3’s N-terminal domain controls the position of LD and FA absorption [3]. ACSL3 grows LD; in the absence of LD, ACSL3 may be present in the ER and translocated to the LD surface during LD formation or after LD is reconnected to ER by membrane bridges [21]. The transfer of ACSL3 from the ER to the LD occurs because of the strong affinity between ACSL3 and LD [22]. Fujimoto et al. found that the addition of an oleic acid medium increased intracellular LD accumulation, which was accompanied by an increase in ACSL3 expression. Meanwhile, with the use of ACSL3 inhibitor triacsin C, the expression of the ACSL3 correspondingly decreased, accompanied by a significant decrease in intracellular LD content. This suggests that ACSL3 plays an important role in the process of ER budding LD growth and maturation after the cellular uptake of exogenous fatty acids [23]. Further research has revealed that the mechanism of action may be that the LD-associated protein Rab18 (low molecular weight GTP-binding protein) interacts with perilipin 2 (PLIN2), a major LD protein, and forms a complex with PLIN2 and ACSL3 [24].

Although the current study on ACSL3 is limited, some evidence shows that ACSL3 is highly relevant to ferroptosis and that ACSL3 expression is correlated with ferroptosis sensitivity [5]. ACSL3 is essential for activating MUFA, which reduces the susceptibility of plasma membrane lipids to oxidation over a period of several hours. MUFA can modify cell membrane properties by replacing PUFAs. It inhibits iron-dependent oxidative cell death by preventing the accumulation of lipid ROS in the plasma membranes, and effectively inhibits the process of iron-dependent oxidative cell death. Based on the findings of Magtanong et al., treatment with exogenous MUFA reduces the sensitization of plasma membrane lipids to lethal oxidation within a few hours, and this process entails the activation of MUFA by ACSL3 [12]. This study demonstrated that exogenous MUFAs and ACSL3 activities promote cellular resistance to ferroptosis. It has also been found that a combination of the anesthetic drug propofol or propofol injectable emulsion (PIE) and the chemotherapeutic drug paclitaxel inhibited cell viability and augmented the sensitivity of paclitaxel to cervical cancer cells. Mechanistically, ferroptosis-related pathways were influenced by drug treatments, including the SLC7A11/GPX4 ubiquinol/CoQ10/FSP1 and YAP/ACSL4/TFRC pathways [25].

### 3.2. ACSL4 in Ferroptosis

As a specific biomarker and driver of ferroptosis, ACSL4 dictates ferroptosis sensitivity by altering the composition of cellular lipids [13,26]. Ferroptosis is induced by the iron-dependent peroxidation of lipids, mostly PLs containing PUFAs [13]. ACSL4 is not only a sensitive monitor for ferroptosis but an essential regulator of lipid metabolism in ferroptosis. The ACSL4 gene-encoded enzyme has a strong preference for 20-carbon PUFA substrates, such as arachidonic acid (AA) and adrenaline (ADA), catalyzing their conversion to AA-CoA and ADA-CoA, producing LPO, and finally promoting ferroptosis [27,28]. Notably, oxidized AA and epinephrine phosphatidylethanolamines (Pes) have been proven to induce ferroptosis in cells [29].

A mechanistic study showed that ACSL4 is a target gene of the oncoprotein YAP. Such an interaction between E-cadherin-mediated cell-cell contacts affecting the cell density on ferroptosis in epithelial cells could activate intracellular Hippo signaling via the NF2/merlin tumor suppressor, and thus inhibit the transcription of various target genes of YAP, including *ACSL4*, *TfR1*, and others [30]. This discovery indicates that additional potential signaling pathways regulating ferroptosis by influencing ACSL4 levels are worthy of investigation and targeting these pathways may be useful in regulating cell death. An analysis revealed that ACSL4 and GPX4 are novel predictive and prognostic biomarkers for patients receiving neoadjuvant chemotherapy, which provided promising evidence for the regulation of ferroptosis in BC treatment [31]. Currently, ACSL4 is studied in cervical cancer mostly in relation to drug therapy. Oleanolic acid (OA) is an anti-cancer compound found naturally in the leaves, fruits, and rhizomes of plants. Research has shown that OA activates ferroptosis in cervical cancer cells by enhancing ACSL4 expression and then a significant decrease in the viability and proliferative capacity of cancer cells was observed after exposure to OA [32]. Curcumin, a yellow polyphenol compound derived from turmeric plants, has also been shown to induce ferroptosis with increasing levels of ACSL4, which presents a theoretical basis for the application of ferroptotic inducers in the treatment of lung cancer [33]. Clinically, radiotherapy upregulates the expression of ACSL4, thereby increasing lipid synthesis and subsequent oxidative damage, thus inducing ferroptosis [34]. In addition to its involvement in tumor progression, ACSL4 serves as a biomarker for sorafenib in the treatment of hepatocellular carcinoma (HCC) [35]. The results of this study demonstrate that the ETS1/miR-23a-3p/ACSL4 axis contributes to sorafenib resistance in HCC by regulating ferroptosis. ACSL4 is upregulated in ovarian cancer (OC) and is a direct target of miR-424-5p, which inhibits ferroptosis in OC cells [36]. The results of these studies have identified a potential therapeutic strategy for treating OC. These studies clearly indicate that there is a relationship between ACSL4 and ferroptosis in various cancers; however, more mechanisms among them remain to be further investigated and may contribute to more effective clinical treatments.

## 4. ACSL3 and ACSL4 in Cancers

Abnormal cell proliferation is an attractive feature of tumors, and the processes of cell membrane formation and abnormal signaling molecules in tumor cells require increased involvement in fatty acid metabolism [37]. Many recent studies have found that ACSLs is closely related to tumorigenesis and development. ACSLs express differently in distinct tumor diseases, and they play an important regulatory role in the proliferation, apoptosis, migration, and invasion of tumor cells; moreover, the expression of ACSLs has a certain influence on the differentiation of tumor tissues, tumor aggression, and prognosis [3].

Studies comparing and exploring the mechanisms of ACSL3 and ACSL4 are relatively limited. An exploration using public databases found that not only do ACSL3 and 4 play a pivotal role in fatty acid metabolism in normal tissues, but they are also frequently expressed at altered levels in cancer, which is sometimes associated with more aggressive metastasis and poor prognosis [38]. The expression of both enzymes was increased in HCC compared with normal liver [39]. Results of the present study demonstrate that analysis of ACSL3 and ACSL4 expression can distinguish between different types of liver tumors. Another study on different types of breast cancers revealed that ACSL3 and ACSL4 were highly, but in a differential manner, expressed in a group of leiomyosarcomas, fibrosarcoma, and rhabdomyosarcomas [40]. A limited number of studies have been conducted on both ACSL3 and ACSL4. The differential expression of these two genes in tumor cells has been observed in some previous studies; however, their effects on either promoting or inhibiting tumor development have not been analyzed in depth.

According to recent studies, ACSL3 is over-expressed in most cancer types and mainly promotes the development of cancer in different ways. The expression and functions of various types of cancers are listed in Table 1. ACSL4 is expressed differently in several cancers and has emerged as an attractive therapeutic target because its inhibition reduces cancer cell invasion and migration. The distinct expressions and functions are summarized in Table 2.

### 4.1. ACSL3 in Various Cancer Types

#### 4.1.1. Breast Cancer (BC)

The study showed that the activity of ACSL3 was suppressed by cell-surface antigen CDCP1 to reduce LD’s abundance and stimulate fatty acid oxidation (FAO). Then FAO contributes to the energy production required for the migration and metastasis of triple-negative breast cancer (TNBC) cells [41]. The findings of this study have important implications for therapeutics targeting TNBC, as well as the development of prognostic markers. The function of ACSL3 in other types of BC remains unclear and requires further exploration.

#### 4.1.2. Colorectal Cancer (CRC)

Regarding CRC, a study demonstrated that ACSL3 was up-regulated by TGF-β1 through sterol regulatory element binding protein 1 (SREBP1) signaling to promote metabolic reprogramming. ACSL3-mediated FAO was required for epithelial-mesenchymal transition (EMT) and metastasis [42]. Targeting the ACSL3 and FAO metabolic pathways may provide therapeutic benefits for CRC.

#### 4.1.3. Gastric Cancer (GC)

A study has shown that ACSL3 was upregulated by MAT2A, which mediated the formation of S-adenosylmethionine (SAM), the main biological methyl donor. Mechanistically, MAT2A increases the trimethylation of lysine-4 on histone H3 (H3K4me3) at the promoter region and subsequently causes ferroptosis resistance [43].

#### 4.1.4. Lung Cancer

A previous study has shown that ACSL3 is robustly expressed at a high level in different lung cancers, including adeno-cell, squamous-cell, large-cell, and small-cell carcinomas. In NSCLC, the overexpression of ACSL3 increases fatty acid activation, leading to more aggressive and invasive cancers and consequently a poorer outcome for patients [45]. In another study, ACSL3 drove prostaglandin synthesis and promoted proliferation and anchorage-independent growth of NSCLC cells [44]. Increased synthesis of prostaglandins is a negative prognostic marker in lung cancer [66], and it can be a potential therapeutic intervention.

#### 4.1.5. Pancreatic Cancer (PDAC)

ACSL3 is overexpressed in PDAC tissues compared to that in normal tissues and correlates with fibrosis, which is linked to poor prognosis and treatment resistance [46]. ACSL3 decreased plasminogen activator inhibitor-1 (PAI-1) secretion from tumor cells and increased tumor fibrosis. PDAC cells are dependent on extracellular fatty acids. This study showed that depletion of extracellularly derived lipids by restriction of ACSL3 could trigger autophagy and reduce PDAC cell proliferation. These results also indicate that the combination of autophagy and ACSL3 inhibition could have better efficacy and survival outcomes than those with single treatments [47].

#### 4.1.6. Prostate Cancer

A previous study showed that OCT1, an androgen receptor (AR) collaborative factor, can coordinate AR signaling to promote the growth of prostate cancer. ACSL3 is a major target of the AR/OCT1 complex, and its high expression is associated with poor prognosis [48]. Additionally, ACSL3 can regulate intratumoral steroidogenesis, thereby contributing to the growth of castration-resistant prostate cancer [49].

### 4.2. ACSL4 in Various Cancer Types

#### 4.2.1. Breast Cancer

According to a previous study, ACSL4 expression was significantly higher in BC tissues than in adjacent normal tissues [50]. ACSL4 is required for cellular uptake of exogenous PUFA and for cancer cells to exhibit a more malignant phenotype [67,68]. Moreover, the expression of ACSL4 may be considered a prognostic indicator and a potential therapeutic target in BC. ACSL4 has also been implicated in drug resistance in breast cancer cell lines through the regulation of ATP-binding cassette (ABC) transporter expression [69]. Researchers have found that ACSL4 could be considered a new therapeutic target to regulate the expression of transporters involved in anti-cancer drug resistance through the mTOR pathway. ACSL4 plays a crucial role in mediating the radio-resistance of breast cancer through its regulation of FOXM1, an extensively studied transcription factor that is associated with the majority of malignancies [70]. Further studies are required to elucidate these mechanisms.

#### 4.2.2. Colorectal Cancer

Studies have identified a pro-tumor metastatic effect of ACSL4 in CRC. ACSL4 is upregulated in CRC and this is associated with a more glycolytic phenotype [51,52]. Other findings have indicated that ACSL4 was upregulated and exhibited a positive correlation with the levels of vascular endothelial growth factor receptor (VEGFR) 1 and 2 [71]. Apatinib, a competitive inhibitor of VEGFR2, has been found to promote ferroptosis by targeting ELOVL6/ACSL4 signaling, supporting the clinical application of apatinib in CRC via a new mechanism [72].

#### 4.2.3. Gastric Cancer

Recent studies suggested that ACSL4 may play a tumor-suppressive role and serve as a therapeutic target in the treatment of GC [53]. A previous study showed that ACSL4 levels are frequently lower in GC tissues, and it inhibited cell growth, colony formation, and migration. The hypoxic microenvironment is a common hallmark of solid tumors and there is growing evidence that hypoxia-inducible factor 1/2α (HIF-1/2α) is involved in tumor treatment resistance, invasion, and metastasis [54]. HIF-1/2α decreased CBS mRNA stability in an RNA, N6-methyladenosine (m6A) dependent manner. CBSLR, which was transactivated by HIF-1α under hypoxic conditions markedly decreased the mRNA stability of CBS and correlated with a poorer prognosis and a poorer response to chemotherapy. Additionally, the hypoxia-induced CBSLR/CBS signaling axis modulates ACSL4 methylation and consequently protects GC cells from ferroptosis [73].

#### 4.2.4. Glioma

In the present study, researchers found that ACSL4 promoted ferroptosis in glioma cells and functioned with anti-proliferative effects [55]. The depletion of ACSL4 reportedly reduces tumor necrosis, proliferation, migration, and cell self-renewal ability [56,57]. In a study that used erastin treatment to increase ACSL4 levels, heat shock protein 90 (Hsp90)–dynamin-related protein 1 (Drp1)–Acsl4 upregulation actively regulates ferroptosis by generating lipid ROS and altering mitochondrial morphology [58]. Therefore, ACSL4 may serve as a novel therapeutic target in glioma treatment.

#### 4.2.5. Gallbladder Cancer

Sirtuin 3 (SIRT3), an NAD^+^-dependent histone deacetylase, is down-regulated in gallbladder cancer and correlates with poor overall survival [59]. The results showed that SIRT3 could inhibit AKT-dependent mitochondrial metabolism, leading to the down-regulating of ACSL4 and the subsequent promotion of ferroptosis. Decreased SIRT3 and ACSL4 levels could also promote EMT markers and invasive activity.

#### 4.2.6. Hepatocellular Cancer

ACSL4 modulates aberrant lipid metabolism and promotes the proliferation and metastasis of HCC cells [60]. Hexokinase 2 (HK2) could activate the transcription of ACSL4, leading to an increase in fatty acid β-oxidation activity which could promote liver cancer growth. In this study, researchers also found that blocking HK2 or ACSL4 effectively inhibited cancer progression, which provides a promising therapeutic strategy for the treatment of liver cancer [61]. Moreover, ACSL4 could mediate the function of METTL5 (the human 18S Rrna m6A methyltransferase) on fatty acid metabolism and targeting ACSL4 and METTL5 synergistically inhibited HCC tumorigenesis [74,75].

#### 4.2.7. Lung Cancer

ACSL4 functions as an anti-tumor agent in lung adenocarcinomas by suppressing tumor survival and invasiveness, and promoting ferroptosis [62]. In an analysis of the TCGA database and validation samples, ACSL4 was frequently downregulated in lung adenocarcinoma, and a lower ACSL4 expression was associated with a poorer prognosis.

#### 4.2.8. Pancreatic Cancer

The expression of ACSL4 was downregulated in PDAC and negatively correlated with the expression of PTPMT1 [63]. High PTPMT1 levels inhibit ferroptosis by suppressing ACSL4 expression, yielding a lower overall survival rate in PDAC.

#### 4.2.9. Prostate Cancer

ACSL4 expression is increased in castration-resistant prostate cancer (CRPC) compared to that in hormone-naive prostate cancer [64] and is involved in the development of tumor aggressiveness in prostate cancer through the regulation of various signal transduction pathways [65]. ACSL4 increased cell proliferation, migration, and invasion by upregulating distinct pathway proteins including p-AKT, lysine-specific demethylase 1 (LSD1), and β-catenin, suggesting that ACSL4 could serve as a biomarker and potential therapeutic target for CRPC.

## 5. ACSL3, ACSL4 and Tumor Immunity

Recent studies have extensively explored the mechanisms that determine susceptibility to ferroptosis in tumor cells; however, studies in immune cells, such as T cells, are scarce.

Research on ACSL3 and the cancer-immune microenvironment has shown that ACSL3 not only increases the fibrosis of PDAC by increasing PAI-1 (a profibrotic TGF-β–responsive gene) secretion from tumor cells but also drives the tumor microenvironment toward immunosuppression, which predicts poor PDAC patient survival [46].

A recent study has confirmed that ACSL4 activates CD8^+^ T cells during ferroptosis. Additionally, studies indicated that CD8^+^ T cells with negative CD36 expression (CD36^−^CD8^+^ T cells) that translocate fatty acids underwent ferroptosis and produced less interferon-gamma in TMEs containing abundant fatty acids and oxidized lipids [76,77,78]. Some evidence has shown that there is an association between CD36 expression and lipid peroxidation and ferroptosis among CD8^+^ T cells infiltrating tumors [79]. CD36-mediated ferroptosis impairs the anti-tumor efficacy of intertumoral CD8^+^ T-cell effectors. This study further found that higher ferroptosis levels in CD8^+^ T cells enhanced their antitumor function and CD36^−^CD8^+^ T cells combined with PD-1 antibodies showed better antitumor effect and survival time, suggesting that targeting ferroptosis is a promising therapy for improving immune checkpoint blockade (ICB)-based tumor immunotherapy. It has been shown that ACSL4 expression is significantly correlated with the amount of CD8^+^T cells present in bladder cancer (BLCA) [80]. Patients with BLCA and elevated ACSL4 levels had better prognoses and suppressed tumor progression. Protein kinase C (PKCs), which control the activity of downstream proteins by phosphorylating their serine/threonine residues, play an important role in various cellular processes, including cell proliferation, survival, and death. Notably, PCKs promoted ROS generation. A study has shown that the PKCβII (one of PKCs isoforms)–ACSL4 pathway modulated ferroptosis and anti-cancer immunity [81].

Additional studies are needed to better understand ferroptosis and anti-cancer immunity and how to target ACSL3 or ACSL4 for better immunotherapeutic efficacy.

## 6. Conclusions

The reprogramming of lipid metabolism in malignant tumors has been identified as a critical feature. An important factor that determines tumor cell survival is the close interaction between ACSL3, ACSL4, and ferroptosis. As discussed in this review, ACSLs such as ACSL3 and ACSL4 have a double-edged effect, that is, they have both anti-tumor as well as pro-tumor effects. Accordingly, the design of inhibitors or activators for ACSL family-specific tumor therapies varies depending on the tumor type. Ferroptosis is an important form of RCD that occurs because of excessive lipid peroxidation caused by iron overload. Additionally, it is becoming increasingly evident that ferroptosis induction has a potential anticancer effect and is regulated by lipid metabolism during cancer initiation and development.

An exploration using public databases found that not only do ACSL3 and 4 play a pivotal role in fatty acid metabolism in normal tissues, but they are also frequently expressed at altered levels in cancer, which is sometimes associated with more aggressive metastasis and poor prognosis [38]. The differential expression of these two genes in tumor cells has been observed in previous studies; however, their effects on either promoting or inhibiting tumor development have not been analyzed in depth [39,40,82]. Moreover, it is unclear how MUFAs competitively affect the oxidation of PUFAs and how MUFA and PUFA affect each other during lipid peroxidation in ferroptosis. It is still necessary to investigate in depth the mechanism by which ACSL3 and 4 participate in the ferroptosis process of tumor cells, as well as the related regulatory factors, in order to develop a better theoretical basis for therapies targeting malignant tumors. The diametrically contrasting effects of ACSL3 and ACSL4 in regulating PUFA and MUFA are not clear, and further studies could investigate the regulatory mechanisms between the two enzymes. This may contribute to the orchestration of ACSL3 and ACSL4, and the development of more efficient and safe targeted drugs.

## Figures and Tables

**Figure 1 cancers-14-05896-f001:**
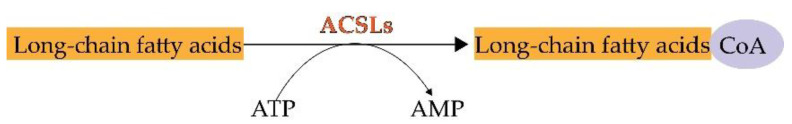
The chemical reaction of ACSLs.

**Figure 2 cancers-14-05896-f002:**
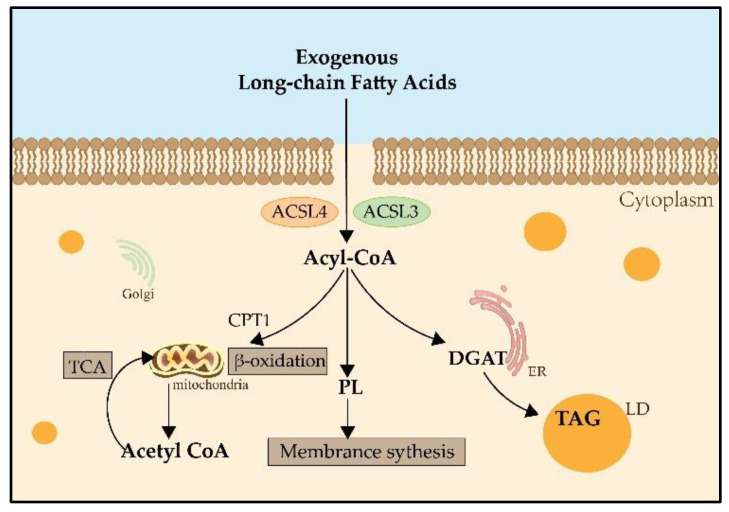
The metabolic pathways of long-chain fatty acids. Both ACSL3 and ACSL4 catalyze exogenous long-chain fatty acids into Acyl-CoA. Acyl-CoA can enter the mitochondria by the transportation of CPT1, then catalyze the formation of Acetyl-CoA and participate in the TCA for providing energy. Acyl-CoA also can produce PLs, which contribute to membrane synthesis. And Acyl-CoA can catalyze DGAT and be involved in the synthesis of TAG in the LDs for lipid storage. CPT1: carnitine palmitoyltransferase 1; TCA: tricarboxylic acid cycle; PLs: phospholipids; DGAT: Diglyceride Acyltransferase; TAG: triglyceride; LD: lipid droplet.

**Figure 3 cancers-14-05896-f003:**
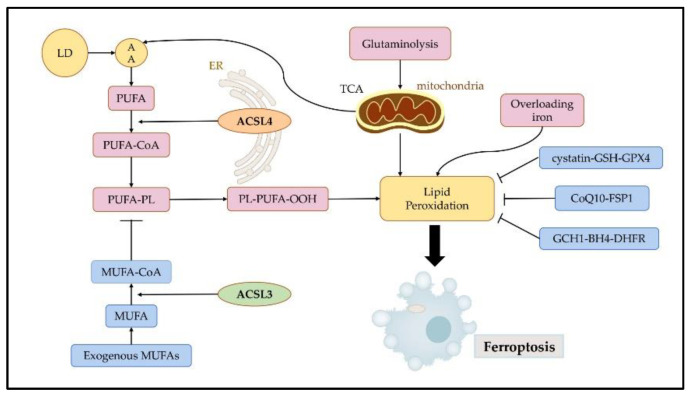
Regulatory mechanisms of ferroptosis. Ferroptosis is mainly initiated by excessive production and failure to eliminate iron, lipid peroxidation, and subsequent plasma membrane rupture. Ferroptosis is induced by lipid peroxidation. ACSL4 contributes to the synthesis of high levels of PUFAs, which promote ferroptosis. ACSL3 is key to MUFA-induced ferroptosis resistance. Three pathways, namely the cystatin-GSH-GPX4, CoQ10-FSP1, and GCH1-BH4-DHFR axis can effectively resist ferroptosis by inhibiting lipid peroxidation. LDs: lipid droplets; AA: arachidonic acid; PUFA: polyunsaturated fatty acids; MUFA: monounsaturated fatty acids; ER: endoplasmic reticulum; TCA: tricarboxylic acid cycle.

**Table 1 cancers-14-05896-t001:** ACSL3 in different cancers.

Cancer	Expression	Functions	Reference
Breast Cancer	↓	CUB-domain-containing protein 1 (CDCP1) promotes cancer cells’ metastasis migration and metastasis by suppressing ACSL3 activity.	[41]
Colorectal Cancer	↑	ACSL3 promotes EMT and metastasis.	[42]
Gastric Cancer	↑	Activation of the Methionine Adenosyltransferase 2A (MAT2A)-ACSL3 pathway drives cells to resist ferroptosis in gastric cancer.	[43]
Lung Cancer	↑	ACSL3 promotes the proliferation and invasiveness of non-small cell lung cancer (NSCLC) cells.	[44,45]
Pancreatic Cancer	↑	ACSL3 increases tumor fibrosis, which is linked to poor prognosis and treatment resistance.ACSL3 could trigger autophagy and reduce pancreatic cancer cell proliferation.	[46,47]
Prostate Cancer	↑	ACSL3 is the major downstream target of AR and (Recombinant Octamer Binding Transcription Factor 1) OCT1 for prostate cancer progression. ACSL3 contributes to the growth of CRPC through intertumoral steroidogenesis	[48,49]

”↑” represents up-regulated expression and “↓” represents down-regulated expression.

**Table 2 cancers-14-05896-t002:** ACSL4 in cancers.

Cancer	Expression	Function	Reference
Breast cancer	↑	High expression of ACSL4 promoted tumor aggression and was a prognostic indicator and potential therapeutic target.	[50]
Colorectal cancer	↑	ACSL4 promoted EMT and metastasis.	[51,52]
Gastric Cancer	↓	ACSL4 inhibited cell growth, colony formation, and migration.	[53,54]
Glioma	↓	ACSL4 promoted tumor necrosis, proliferation, migration, and cell self-renewal ability.	[55,56,57,58]
Gallbladder Cancer	↓	SIRT3 inhibited ACSL4 expression that drives ferroptosis and promotes the activity of epithelial-mesenchymal (EMT) markers and invasiveness	[59]
Hepatocellular Cancer	↑	ACSL4 promoted the proliferation and metastasis of HCC cells.	[60,61]
Lung Cancer	↓	ACSL4 suppressed tumor survival and invasiveness and promoted ferroptosis.	[62]
Ovarian cancer	↑	ACSL4 was a direct target of miR-424-5p which showed inhibition on ferroptosis that serves as a novel tumor suppressor.	[36]
Pancreatic Cancer	↓	High protein tyrosine phosphatase mitochondria1(PTPMT1) level inhibited ferroptosis by suppressing the expression of ACSL4 and displayed a lower overall survival rate.	[63]
Prostate cancer	↑	ACSL4 increased cell proliferation, migration, and invasion.	[64,65]

”↑” represents up-regulated expression and “↓” represents down-regulated expression.

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
