# Peer review of "ACSL3 and ACSL4, Distinct Roles in Ferroptosis and Cancers"

_cancers, 2022, doi:10.3390/cancers14235896_

Round 1
Reviewer 1 Report
Authors reviewed the heterogeneity and functions of ACSL3 and ACSL4 in ferroptosis, and provide the current understanding of and research on ACSL3 and ACSL4 in multiple cancers.
This review is very interesting to readers of this journal. My comments are as follow.
In Introduction Section, authors described that ACSL4 is a positive regulator in ferroptosis. However, authors described that ACL4 inhibition reduces cancer cells invasion and migration, and induces apoptosis and ferroptosis in section 3.2. The opposite is shown as a function of ASCL4 against ferroptosis, which creates confusion. CDCP1 promotes TNBC metastasis migration and metastasis by suppressing ACSL3, whereas low ACSL3 expression correlates strongly with sensitivity to ferroptosis-inducing agents in hundreds of cancer cell lines. The former described anti-tumor effects, while the latter described pro-tumor effects. Finally, I was able to understand that ACSL3 and ACSL4 have a double-edged effect, that is, they have both anti-tumor as well as pro-tumor effects. Therefore, I would like to ask you to rethink the structure of your paper. Specifically, it would be easier to understand if you bring the effects of ACSL3 and ACSL4 on ferroptosis in the first half of the paper and discuss the functions of ACSL3 and ACSL4 in various cancer types in the second half.
It should be mentioned that in Section 3.1 Table 1 summarized the expression and function of ACL3 in various cancer types; the same applies to ACL4.
Ovarian cancer is mentioned in Table 2 but not in the text.
In page 2 and 3, FAO is “fatty acid oxidation”?
What is PIE in page 7?
Author Response
Response to Reviewer 1 Comments:
Thank you for your nice comments on our article. According to your suggestions, we have supplemented several contents here and corrected several mistakes in our previous draft. Based on your comments, we also attached a point-by-point letter to you. We have made extensive revisions to our previous draft. The detailed point-by-point responses are listed below.We have also attached the document in the end.Please see the attachment.
Point 1: In Introduction Section, authors described that ACSL4 is a positive regulator in ferroptosis. However, authors described that ACL4 inhibition reduces cancer cells invasion and migration, and induces apoptosis and ferroptosis in section 3.2. The opposite is shown as a function of ASCL4 against ferroptosis, which creates confusion.
Response 1: We agree with the comments and apologize for our misleading and wrong descriptions.ACSL4 is a positive regulator in ferroptosis by altering the composition of cellular lipids or activating signaling pathways. ACSL4 have different effects in different cancers. ACSL4 is up-regulated in some types of cancers such as breast cancerglioma and so on and can promote cancer cells invasion and migration. But ACSL4 inhibites cell growth, colony formation, and migration in gastric cancer.
Point 2: CDCP1 promotes TNBC metastasis migration and metastasis by suppressing ACSL3, whereas low ACSL3 expression correlates strongly with sensitivity to ferroptosis-inducing agents in hundreds of cancer cell lines. The former described anti-tumor effects, while the latter described pro-tumor effects.
Response 2: Thank you for pointing this out. ACSL3 has a double-edged effect. On one hand, ACSL3 activities promote cellular resistance to ferroptosis. On the other hand, ACSL3 has anti-tumor effects in TNBC because it is suppressed by CUB-domain-containing protein 1 (CDCP1), stimulating fatty acid oxidation (FAO). FAO can produce energy required for the migration and metastasis of triple-negative breast cancer (TNBC) cells.
Point 3: Finally, I was able to understand that ACSL3 and ACSL4 have a double-edged effect, that is, they have both anti-tumor as well as pro-tumor effects. Therefore, I would like to ask you to rethink the structure of your paper. Specifically, it would be easier to understand if you bring the effects of ACSL3 and ACSL4 on ferroptosis in the first half of the paper and discuss the functions of ACSL3 and ACSL4 in various cancer types in the second half.
Response 3: We deeply appreciate your suggestion and your comments are absolutely right. We have adjusted the structure of our paper to make it easier to understand. We have put “ACSL3 and 4 in ferroptosis” in the first half and discuss their roles in different cancers. Your suggestion is so important and useful to our paper. Please check our structure in our revised manuscript.
Point 4: It should be mentioned that in Section 3.1 Table 1 summarized the expression and function of ACL3 in various cancer types; the same applies to ACL4.
Response 4: We have added the summaries of the expression and function of ACSL3 and ACSL4 in various cancer types.
Point 5: Ovarian cancer is mentioned in Table 2 but not in the text.
Response 5: The part about the effects of ACSL4 in ovarian cancer is mentioned in the section “ACSL4 in ferroptosis” as the following: ACSL4 is upregulated in ovarian cancer (OC) and is a direct target of miR-424-5p, which inhibits on ferroptosis in OC cells[36]. We didn’t add the contents in the section” ACSL4 in cancers” to avoid repetitions.
Point 6: In page 2 and 3, FAO is “fatty acid oxidation”?
Response 6: Thanks for pointing out this problem. FAO means fatty acid oxidation and we have corrected it in the article.
Point 7: What is PIE in page 7?
Response 7: PIE represents propofol injectable emulsion and is repeated with propofol. Thank you for your comments and we have added it in the article as the following “It has also been found that combination of the anesthetic drug propofol or propofol injectable emulsion (PIE) and the chemotherapeutic drug paclitaxel inhibited cell viability and augmented the sensitivity of paclitaxel to cervical cancer cells.”.

Reviewer 2 Report
The manuscript by Yufei Yang et al described the roles of ACSL3 and ACSL4 and how they are related in ferroptosis in cancer. They summarized the roles of ACSL3 and 4 in each cancer and some of the mechanisms of ACSL3 and 4 in ferroptosis.
Major Comments
In section 2, they described briefly about the reaction of ACSL3 and 4. But it is not clear. It is better to add a figure to explain what is the chemical reaction of ACSL3 and 4. In addition, In the later section 4, the authors described that ACSL3 and 4 have different roles. It is not clear what is the common reaction of ACSL3 and 4, and what is the specific reaction that produces the different roles in each cancer. This point should be clarified in the manuscript.
Minor comments
In Figure 1, it is not clear why ferroptosis occur by lipid peroxidation. Probably better to add cell membrane rupture in the figure.
In section 5, CD36-CD8 T cells should be explained what it means to the readers.
In the end of introduction, “Furthermore, we present a perspective on cancer therapies targeting or combining. ACSL3 and ACSL4 in ferroptosis.” Probably period should be removed between combining and ACSL3.
In 3.1.1, FAO should be fully spelled as it first appears. In 3.1.2, the word “fatty acid oxidation” come out. Probably this is FAO. Also, in 3.1.4, the word “fatty acid activation” appears. It is not clear what is the reaction of “fatty acid activation”.
In the legend of Figure 1, “Ferroptosis is induced by lipid peroxidation” is repetitive to the previous sentence.
4.1 “Oleanolic acid (OA) is an anti-cancer compound found naturally in in the leaves, fruits, and rhizomes of plants.” “in” is repetition.
5. “Notably, PCKs promoted ROS generation.” Probably, PCKs is PKCs.
Author Response
Response to Reviewer 2 Comments
Thank you for your nice comments on our article. According to your suggestions, we have supplemented several contents here and corrected several mistakes in our previous draft. Based on your comments, we also attached a point-by-point letter to you. We have made extensive revisions to our previous draft. The detailed point-by-point responses are listed below.We have also attached the document in the end.
Point 1: In section 2, they described briefly about the reaction of ACSL3 and 4. But it is not clear. It is better to add a figure to explain what is the chemical reaction of ACSL3 and 4. In addition, In the later section 4, the authors described that ACSL3 and 4 have different roles. It is not clear what is the common reaction of ACSL3 and 4, and what is the specific reaction that produces the different roles in each cancer. This point should be clarified in the manuscript.
Response 1: We are so grateful for the suggestion because it is important to show clear and visual explanations. To be more clearly and in accordance with your comments, we have added a more detailed figure in the section 2 to explain the common reactions of ACSL3 and 4. The figure are shown in the attached Word and the revised manuscript.
Point 2: In Figure 1, it is not clear why ferroptosis occur by lipid peroxidation. Probably better to add cell membrane rupture in the figure.
Response 2: It was our mistake for ignoring the cell membrane rupture induced by lipid peroxidation. We have modified my drawing in the review. Thanks for raising this important issue.
Point 3: In section 5, CD36-CD8 T cells should be explained what it means to the readers.
Response 3: Thank you for raising this issue. We have changes “CD36-CD8 T cells” into “CD8+ T cells with negative CD36 expression (CD36-CD8+ T cells)”.
Point 4: In the end of introduction, “Furthermore, we present a perspective on cancer therapies targeting or combining. ACSL3 and ACSL4 in ferroptosis.” Probably period should be removed between combining and ACSL3.
Response 4: We apologize for the language problems in the original manuscript. The “.” is extra and I have deleted it.
Point 5: In 3.1.1, FAO should be fully spelled as it first appears. In 3.1.2, the word “fatty acid oxidation” come out. Probably this is FAO. Also, in 3.1.4, the word “fatty acid activation” appears. It is not clear what is the reaction of “fatty acid activation”.
Response 5: Thanks for pointing out this problem. FAO means fatty acid oxidation and we have corrected it in the article.
Point 6: In the legend of Figure 1, “Ferroptosis is induced by lipid peroxidation” is repetitive to the previous sentence.
Response 6: Thanks for pointing out this problem. This sentence is repeated and we have deleted it.
Point 7: 4.1 “Oleanolic acid (OA) is an anti-cancer compound found naturally in in the leaves, fruits, and rhizomes of plants.” “in” is repetition.
Response 7: We were really sorry for our careless mistakes. Thank you for your reminder. The correct statement is very important and we have removed the repeated “in”.
Point 8: 5. “Notably, PCKs promoted ROS generation.” Probably, PCKs is PKCs.
Response 8: We feel sorry for our carelessness. In our resubmitted manuscript, the PKCs has been revised. Thanks for your correction.

Reviewer 3 Report
In the manuscript, the authors attempted to determine the correlation between ACSL3 and ACSL4 and cancer promotion or inhibition in different cancer types. Also, the role of ACSL3 and ACSL4 in ferroptosis regulation as a type of cell death. However, while a few recent reviews explained the relationship between ACSLs and cancer via ferroptosis, the limitations of both ACSL3 and ACSL4 were not discussed in depth. Also, the relation between ACSL3 and ACSL4 and tumor immunity is very interesting, although it is very short. The manuscript is well written, and the researchers explain the outcomes of the review very well. Most references were recent. The illustration is simple, and the tables were clear and probably describe the regulation of ACSL3 and ACSL4 in different cancers. The conclusion summarized the entire review and suggested additional research in that field. I would like to suggest some changes that should be made to clarify the review for the readers.
- The abbreviations section must be included.
- The authors write the abbreviation ACSLs as "long-chain fatty acyl CoA synthetase", but it must be acyl-CoA synthetase long-chain family.
- The author stated, "In addition, ACSL4 is a positive regulator in ferroptosis, whereas ACSL3 inhibits ferroptosis", but no reference was added to confirm this sentence.
- In 3.1.1. Breast Cancer (BC), "………. ACSL3 was suppressed to reduce lipid droplets (LD) abundance and stimulating FAO…." What is the FAO? The author must explain it. Many abbreviations need to be clarified.
- There is nothing about Table 1 and Table 2 within the text.
- In 3.2.4. Glioma "In the present study, we observed that ferroptosis in human glioma tissues and glioma cells was reduced and the expression of ACSL4 was also downregulated[34]" Please revise this sentence as it is a review article, and reference 34 didn’t belong to the authors.
- "Although the current study on ACSL3 is limited, some evidence shows that ACSL3 is highly relevant to ferroptosis and that ACSL3 expression is correlated with ferroptosis sensitivity." References are needed to confirm this information.
- "Moreover, OA activates ferroptosis in HeLa cells by enhancing ACSL4 expression [62]." I think this part should be transferred to the next section 4.2. ACSL4 in Ferroptosis.
- Few studies that correlate the ACSL3 and ACSL4 with various cancers are missed in this review and presented in other reviews, authors must add all the studies in this review to benefit the reader.
Author Response
Response to Reviewer 3 Comments:
Thank you for your nice comments on our article. According to your suggestions, we have supplemented several contents here and corrected several mistakes in our previous draft. Based on your comments, we also attached a point-by-point letter to you. We have made extensive revisions to our previous draft. The detailed point-by-point responses are listed below.We have also attached the document in the end.Please see the attachment.
Point 1: The abbreviations section must be included.
Response 1: Thanks for pointing out the details. We have supplemented all abbreviations in the end of the paper as the following:
Abbreviations: AA: Arachidonic Acid; ABC: ATP-Binding Cassette; ACSLs: The Long-Chain Fatty Acyl CoA Synthetase Family; ADA: Adrenaline; AR: Androgen Receptor; BLCA: Bladder Cancer; CD36-CD8+ T cells: CD8+ T Cells With Negative CD36 Expression; CDCP1: Cub-Domain-Containing Protein 1; CRC: Colorectal Cancer; CRPC: Castration-Resistant Prostate Cancer; EMT: Epithelial-Mesenchymal Transition; ER: Endoplasmic Reticulum; FAO: Fatty Acid Oxidation; GC: Gastric Cancer; HCC: Hepatocellular Carcinoma; HIF-1/2α: Hypoxia-Inducible Factor 1/2α; HK2: Hexokinase 2; Hsp90: Heat Shock Protein 90; ICB: Immune Checkpoint Block-ade; LD: Lipid Droplet; LSD1: Lysine Specific Demethylase 1; MUFA: Monounsaturated Fatty Acids; NSCLC: Non‐Small Cell Lung Cancer; OA: Oleanolic Acid; OC: Ovarian Cancer; PAI-1: Plasminogen Activator Inhibitor–1; PDAC: Pancreatic Ductal Adenocarcinoma; PEs: Phosphati-dylethanolamines; PKCs: Protein Kinase C; PLIN2: Perilipin 2; PLs: Phospholipids; PTPMT1: Protein Tyrosine Phosphatase Mitochondria1PUFA: Polyunsaturated Fatty Acids; RCD: Regu-lated Cell Death; ROS: Reactive Oxygen Species; SCD-1: Stearyl Coenzyme A Dehydrogenase-1; SIRT3: Sirtuin 3; SREBP1: Sterol Regulatory Element Binding Protein 1; TAG: Triglyceride; TNBC: Triple-Negative Breast Cancer; VEGFR: Vascular Endothelial Growth Factor Receptor; LPO: Li-pid Peroxides.
Point 2: The authors write the abbreviation ACSLs as "long-chain fatty acyl CoA synthetase", but it must be acyl-CoA synthetase long-chain family.
Response 2: Thank you for raising this issue. It was our mistake and we have changed “long-chain fatty acyl CoA synthetase” into “acyl-CoA synthetase long-chain family” to represent the abbreviation ACSLs.
Point 3: The author stated, "In addition, ACSL4 is a positive regulator in ferroptosis, whereas ACSL3 inhibits ferroptosis", but no reference was added to confirm this sentence.
Response 3: Thanks for raising this important issue. we didn't add a quotation. Therefore, we have rewritten In addition, ACSL4 is a positive regulator in ferroptosis, whereas ACSL3 helps cancer cells acquire ferroptosis resistance[Li D, Li Y. The interaction between ferroptosis and lipid metabolism in cancer. Signal transduction and targeted therapy. 2020;5(1):108. doi: 10.1038/s41392-020-00216-5.].
Point 4: In 3.1.1. Breast Cancer (BC), "………. ACSL3 was suppressed to reduce lipid droplets (LD) abundance and stimulating FAO…." What is the FAO? The author must explain it. Many abbreviations need to be clarified.
Response 4: Thanks for pointing out this problem. We apologize for the language problems in the original manuscript. FAO means fatty acid oxidation and I have corrected it in the article.
Point 5: There is nothing about Table 1 and Table 2 within the text.
Response 5: Thank you for raising this issue, we have added the summaries of the expression and function of ACSL3 and ACSL4 in various cancer types.
Point 6: In 3.2.4. Glioma "In the present study, we observed that ferroptosis in human glioma tissues and glioma cells was reduced and the expression of ACSL4 was also downregulated[34]" Please revise this sentence as it is a review article, and reference 34 didn’t belong to the authors.
Response 6: Thanks for pointing out this problem. We agree with the comment and re-wrote the sentence in the revised manuscript as the following: In the present study, researchers found that ACSL4 promoted the ferroptosis in glioma cells and functioned anti-proliferative effects.
Point 7: "Although the current study on ACSL3 is limited, some evidence shows that ACSL3 is highly relevant to ferroptosis and that ACSL3 expression is correlated with ferroptosis sensitivity." References are needed to confirm this information.
Response 7: Thanks for comments. We have added the reference “5. Li D, Li Y. The interaction between ferroptosis and lipid metabolism in cancer. Signal transduction and targeted therapy. 2020;5(1):108. doi: 10.1038/s41392-020-00216-5.” in the end of the sentence “Although the current study on ACSL3 is limited, some evidence shows that ACSL3 is highly relevant to ferroptosis and that ACSL3 expression is correlated with ferroptosis sensitivity”.
Point 8: "Moreover, OA activates ferroptosis in HeLa cells by enhancing ACSL4 expression [62]." I think this part should be transferred to the next section 4.2. ACSL4 in Ferroptosis.
Response 8: Thank you for raising this insight question. This part has been moved to section 4.2. ACSL4 in Ferroptosis.
Point 9: Few studies that correlate the ACSL3 and ACSL4 with various cancers are missed in this review and presented in other reviews, authors must add all the studies in this review to benefit the reader.
Response 9: We deeply appreciate the reviewer’s suggestion. We have checked the literature carefully and added more references. We also adjusted some parts and orders to make ie easier to read. According to the reviewer’s comment, we have provided more details for studies that correlate the ACSL3 and ACSL4 with various cancers in the section “ACSL3 and ACSL4 in cancers “ as the following:
Studies comparing and exploring the mechanisms of ACSL3 and ACSL4 are relatively limited. An exploration using public databases found that not only do ACSL3 and 4 play a pivotal role in fatty acid metabolism in normal tissues, but they are also frequently expressed at altered levels in cancer, which is sometimes associated with more aggressive metastasis and poor prognosis[12]. Expression of both enzymes was increased in hepatocellular carcinoma (HCC) compared with normal liver[13]. Results of the present study demonstrate that analysis of ACSL3 and ACSL4 expression can distinguish between different types of liver tumors Another study on different types of breast cancers revealed that ACSL3 and ACSL4 were highly, but in a differential manner, expressed in a group of leiomyosarcomas, fibrosarcoma and rhabdomyosar-comas[14]. A limited number of studies have been conducted on both ACSL3 and ACSL4. The differential expression of these two genes in tumor cells has been observed in some previous studies, however, their effects on either promoting or inhibiting tumor development have not been analyzed in depth.
